# Functional Characterization of *MdTAC1a* Gene Related to Branch Angle in Apple (*Malus* x *domestica* Borkh.)

**DOI:** 10.3390/ijms23031870

**Published:** 2022-02-07

**Authors:** Yongzhou Li, Xu Tan, Jing Guo, Enyue Hu, Qi Pan, Yuan Zhao, Yu Chu, Yuandi Zhu

**Affiliations:** Department of Pomology, College of Horticulture, China Agricultural University, Yuanmingyuan West Road No. 2, Haidian District, Beijing 100193, China; yongzhoulee@cau.edu.cn (Y.L.); SY20203172714@cau.edu.cn (X.T.); 18702225153@163.com (J.G.); 18854806315@163.com (E.H.); panlor1204@163.com (Q.P.); zhaoyuan0605@cau.edu.cn (Y.Z.); chuyu818@163.com (Y.C.)

**Keywords:** *Malus* x *domestica* Borkh., *MdTAC1*, branch angle, gene interaction, InDel marker

## Abstract

The Tiller Angle Control 1 (*TAC1*) gene belongs to the *IGT* family, which mainly controls plant branch angle, thereby affecting plant form. Two members of *MdTAC1* are identified in apple; the regulation of apple branch angle by *MdTAC1* is still unclear. In this study, a subcellular localization analysis detected *MdTAC1a* in the nucleus and cell membrane, but *MdTAC1b* was detected in the cell membrane. Transgenic tobacco by overexpression of *MdTAC1a* or *MdTAC1b* showed enlarged leaf angles, the upregulation of several genes, such as GA 2-oxidase (GA2ox), and a sensitive response to light and gravity. According to a qRT-PCR analysis, *MdTAC1a* and *MdTAC1b* were strongly expressed in shoot tips and vegetative buds of weeping cultivars but were weakly expressed in columnar cultivars. In the *MdTAC1a* promoter, there were losses of 2 bp in spur cultivars and 6 bp in weeping cultivar compared with standard and columnar cultivars. An InDel marker specific to the *MdTAC1a* promoter was developed to distinguish apple cultivars and F_1_ progeny. We identified a protein, MdSRC2, that interacts with MdTAC1a, whose encoding gene which was highly expressed in trees with large branch angles. Our results indicate that differences in the *MdTAC1a* promoter are major contributors to branch-angle variation in apple, and the *MdTAC1a* interacts with *MdSRC2* to affect this trait.

## 1. Introduction

Apple (*Malus* x *domestica* Borkh.) is widely cultivated and ranks third with respect to global fruit production according to FAO statistics in 2019 (https://www.fao.org (accessed on 5 November 2020)). On the basis of fruiting type and branch architecture, apple trees are classified into four ideotypes—columnar, standard, spur, and weeping, with different branch angles [1]. Fruit tree architecture determines the compactness of the canopy structure, the amount of shadowing between branches, and the extent of ventilation and light transmission to the lower part of the tree. Branching can affect tree shape, in turn influencing fruit yield and quality [2,3,4,5]. In apple production, branch angles are manually widened to promote early flowering and fruiting. Investigation of branch angle-related genes should therefore contribute to apple tree breeding and genetic improvement [1].

Genes determined to shape plant branch angle include Tiller Angle Control 1 (*TAC1*), *LAZY1* (*LA1*), Loose Plant Architecture 1 (*LPA1*), and Prostrate Growth 1 (*PROG1*) [6]. Branch angle is also affected by growth hormones, especially auxins (e.g., indole acetic acid [IAA]) [6,7], cytokinins [8,9], and strigolactones [10]. The polar transport of auxin inhibits the growth of lateral buds but causes apical buds to grow vigorously. Cytokinins are produced in roots, and their upward transportation promotes lateral bud differentiation and growth, which work in synergy with auxins, thereby affecting branch development. Strigolactones are a recently discovered class of hormones commonly found in plants. Strigolactones can interact with auxins and cytokinins to reduce the extent of plant branching and play an important role in branch development [10]. Many environmental signals, including light, gravity, and thigmotropism, can influence plant architecture as well [11,12]. *TAC1* is a gene first isolated in rice by map-based cloning and is a positive regulator of the tillering angle in rice [13]. Previous research has revealed that *TAC1* originated from *LA1*, which negatively regulates the tillering angle in plants. *TAC1* and *LA1* belong to two branches of the *IGT* gene family, which has been found in all plant genomes and contains a conserved EAR (LxLxL) motif at the C terminus [13].

*TAC1* plays an important role in regulating branch angle in many plant species [13,14,15]. For example, *OsTAC1* participates in the regulation of rice (*Oryza sativa*) tiller angle and influences the endogenous auxin content [16]. In maize (*Zea mays*), the expression level of *ZmTAC1* is positively correlated with leaf angle, and this gene plays an important regulatory role in leaf development [15]. At each stage of wheat (*Triticum aestivum*) tillering, the expression of *TaTAC1* is positively correlated with tiller angle, and this gene may positively regulate tiller angle by participating in the regulation of auxin polar transport [17]. *PpeTAC1* promotes the horizontal growth of branches in peach (*Prunus persica*) trees [14,18], and *PzTAC* contributes to the regulation of branch angle in poplar (*Populus* × *zhaiguanheibaiyang*) [19].

In a previous investigation, we separately cloned the *TAC1a/b* genes of apple (GenBank accession No. MG837476/MG837477). There was no difference in the cDNA sequences, but variation in the promoters of two genes were detected among the analyzed ideotypes [20]. This preliminary work involved few apple cultivars, and a correlation with branch angle could not be determined. In the present study, we performed subcellular localization experiments and used full-length *MdTAC1a/b* cDNA sequences to construct overexpression vectors of *MdTAC1a/b* driven by a strong promoter, CaMV35S, in genetically transformed tobacco (*Nicotiana *benthamiana)**. This approach allowed us to observe changes in the leaf angle of the transgenic lines and verify the gene functions of *MdTAC1a/b*. We then developed new molecular marker based on differences in the 2000-bp long *MdTAC1a* promoters among apple cultivars and a hybrid F_1_ population. Finally, we used yeast two-hybrid technology to study the *MdTAC1a*-interacting protein. Our findings, including the function of *MdTAC1a/b* genes, lay a foundation for analysis of the molecular mechanism of *MdTAC1a* regulation of branch angle in apple, thereby contributing to apple genetic improvement.

## 2. Results

### 2.1. Subcellular Localization Analysis of MdTAC1a and MdTAC1b

To better understand the molecular function of the *MdTAC1*, we constructed *MdTAC1a-GFP* and *MdTAC1b-GFP* fusion gene-expression vectors to examine their subcellular locations. The *MdTAC1a-GFP* and *MdTAC1b-GFP* plasmids were transferred into *Agrobacterium* to infect tobacco leaves. The *MdTAC1a* was detected in the cell nucleus and cell membrane, whereas the *MdTAC1b* was localized to the cell membrane (Figure 1).

### 2.2. Phenotypic and Key Gene Analysis of Transgenic MdTAC1a/b in Tobacco

To investigate the physiological function of *MdTAC1*, we separately transferred *MdTAC1a/b* into tobacco. In a real-time PCR assay, the expressions of *MdTAC1a/b* were detected in *MdTAC1a/b*-overexpressing (OE) plants but not in wild-type (WT) plants (Figure 2). The order of relative expression of three selected *MdTAC1a*-OE transgenic tobacco plants was L1 > L2 > L3, and the relative expression level of *MdTAC1a* in L1 was almost 80 times that of the wild type (Figure 2). The relative expression levels of three selected *MdTAC1b*-OE transgenic tobaccos followed the order L1 > L3 > L2 (Figure 2). In addition, the leaf angle of *MdTAC1a*/*b*-OE transgenic tobacco plants was significantly larger than that of the wild type (Figure 2 and Appendix A).

To compare differences in hormone-expression levels between genetically modified and wild-type tobaccos, we used qRT-PCR to analyze ten genes involved in the synthesis and signal transduction of four types of plant hormones and two genes in the light-responsive pathway after flowering. The relative expression levels of ten of these genes were found to be significantly different between wild-type and transgenic tobaccos (Figure 3). In particular, genes encoding auxin response factors 2 and 3 (*ARF2* and *ARF3*), phototropins *PHOT1* and *PHOT2*, GA2 oxidase (*GA2ox*), carotenoid cleavage dioxygenase 7 (*CCD7*), and *IAA5* were upregulated in genetically modified tobacco compared with the wild type, whereas pin-formed 1 (*PIN1*), pin-formed 2 (*PIN2*), GA20 oxidase (*GA20ox*), and isopentenyl transferase (*IPT*) genes were upregulated in wild-type plants relative to transgenic tobacco. After wild-type and transgenic tobacco plants began to bend in the horizontal treatment, we selected the bending point and measured the relative expression of *IAA5*. We found that *IAA5* expression was significantly higher in transgenic tobaccos than in the wild type.

### 2.3. Phototropism and Gravitropism of MdTAC1-OE Plants

To investigate the role of *MdTAC1a* and *MdTAC1b* in shoot gravitropism and light response, we subjected WT and transgenic plants to 90° inverted gravity processing under light and dark conditions to observe the degree of stem bending. WT and OE plants under light exposure were arranged horizontally for 0, 1, 5, 12, and 24 h for image collection (Figure 4A), and the angle of the stem was measured from the image. The WT plants began to bend upward after 1 h, with significant upward bending observed from 1 to 3 h. After 1 h of treatment, a significant difference in bending angle was observed between OE and WT plants (Figure 4C). In the OE plants, no bending was observed in the first 6 h. After 6 h, the OE plants began to bend gradually, and the rate of bending of *MdTAC1a*-OE plants was greater than that of *MdTAC1b*-OE (Figure 4C).

After 0, 1, 8, 12, 24, and 48 h of horizontal placement in darkness, plants were imaged, and the stem angle was measured. The time required for WT and OE plants to bend upward in the dark was later than that of corresponding plants exposed to light, indicating the transgenic plant response was more sensitive to light and gravity (Figure 4B). During the first 3 h in the dark, no stem bending was observed in WT or OE plants. WT plants began to bend slightly upward after 3 h, whereas *MdTAC1a*-OE and *MdTAC1b*-OE plants began bending slightly upward after 8 h (Figure 4B). The bending rate of WT plants was significantly higher than that of the slowest plants, namely, *MdTAC1b*-OE plants. In addition, the bending rate of *MdTAC1a*-OE was higher than *MdTAC1b*-OE plants under light, whereas the bending rate of *MdTAC1b*-OE plants was significantly higher than that of *MdTAC1a*-OE in the dark (Figure 4D). In addition, the stems were sampled at 0, 1, 5, 12, and 24 h lightening condition and 0, 1, 8, 12, 24, and 48 h dark condition for expression analysis, and the qRT-PCR results showed that the expression levels of *MdTAC1a* and *MdTAC1b* increased significantly under lightening and no significant changes were observed under darkness (Figure 4F). We also measured the expression of IAA5 in the stem bends of WT plants and OE plants at 24 h in the light and 48 h in the dark, and it was clear that the expression of *NbIAA5* was higher than that in WT under light or dark treatment (Figure 4E). We hypothesize that light-response and cis-acting modulator-of-photoreactivity elements in *MdTAC1a* and *MdTAC1b* play a significant role in stem bending in light and gravity (Appendix A).

### 2.4. qRT-PCR Analysis of MdTAC1a/b Expressions in Different Apple Cultivars

Eleven apple cultivars were analyzed for the expressions of *MdTAC1a* and *MdTAC1b.* The branch angle size showed an increasing trend in the order of columnar, standard, spur, and weeping apple cultivars (Appendix A, Figure 5A). The qRT-PCR revealed that the expression levels of *MdTAC1a* and *MdTAC1b* were similar in vegetative buds and shoot tips. The highest expression levels were detected in weeping-type cultivars, followed by standard cultivars, whereas the lowest levels were observed in columnar ones. In addition, the relative expression level of *MdTAC1b* in vegetative buds and shoot tips was higher than that of *MdTAC1a* in these tissues (Figure 5B,C).

### 2.5. Development of an InDel Marker and Cosegregation in a Population

To explore the variation in the promoters of *MdTAC1a/b* genes in the 11 apple tree cultivars, we sequenced the coding region and 2.0-kb region upstream of ATG (position 0) of all 11 cultivars. InDel in the *MdTAC1a* promoter sequence were detected among the four types of cultivars (Figure 6A). A deletion of six nucleotides, from −270 to −275 (GAGAGA), was detected in ‘Granny Smith’, a weeping-type cultivar, while a two-nucleotide deletion (GA) at positions −271 to −270 was identified in the three spur-type cultivars, which belong to the GAG motif. SNPs were observed in *MdTAC1b* in the four cultivar types (Figure 6B). For instance, the deletion of a single C at position −158 was observed in columnar-type apple cultivars. As another example, the A nucleotide present at positions −848, −804, and −552 in standard-type cultivars was a G in all other cultivars. The A at position −724 was replaced by a G in columnar-type cultivars, and the G at position −599 was replaced by an A in apple trees with weeping branches. Columnar- and spur-type cultivars possessed a T and a C at positions −566 and −294, respectively, whereas a C and a G were present at these corresponding positions in all other cultivars. The nucleotide present at position −47 was a C in standard-type cultivars and a T in the remaining cultivars.

As demonstrated by these examples, *MdTAC1a/b* obviously differed among the apple tree ideotypes. We therefore designed a molecular-marker pair to generate a 370-bp long amplification product based on the InDel in the *MdTAC1a* promoter (Appendix A). To test the applicability of this molecular marker to other apple trees, we selected 21 cultivars with different branch angles from the resource nursery. The width of the base-branch angle ranged from 50 cm to 120 cm in the cultivated cultivars and F_1_ generation. Among the 21 cultivars, the branch angle of 4 was ≤45°, that of 7 was 45–65°, and 10 cultivars had a branch angle ≥ 65° (Figure 7). Among the 21 cultivars, PCR amplification yielded the complete 370 bp in 14 cultivars, whereas a 6 bp deletion was found in 7 cultivars (Figure 7). This genotypic ratio was different from the phenotypic statistics, with the phenotype and genotype coinciding in only 85.71% of cultivars.

In addition, 62 F_1_ hybrid offsprings were selected from a cross between the two cultivars, ‘Jinlei No. 1’ and ‘Granny Smith’ to verify the InDel Maker (Figure 8A). Trait segregation in the F_1_ population was observed: 30 plants had a branch angle ≤ 45°, whereas the remaining 32 plants had a branch angle > 45° (Figure 8B). As to genotype, 28 and 34 trees had the same genotype as the parents, ‘Jinlei No. 1’ and ‘Granny Smith’, respectively (Figure 8C), which corresponded to segregation rates of 45.16% and 54.84%. In the F_1_ generation, the phenotype–genotype coincidence rate was 96.77%. The primer pair developed in this study can therefore be applied (Figure 8C).

### 2.6. Interaction of MdTAC1a with MdSRC2

The promoter of *MdTAC1a* has InDel in different cultivars and *MdTAC1a* localized in both the cell membrane and the nucleus; MdTAC1a was selected for yeast two-hybrid in this experiment. Using the *MdTAC1a* as bait, we screened the cDNA interaction library of apple by the yeast two-hybrid method. After identifying the candidate gene, *SOYBEAN GENE REGULATED BY COLD 2* (*MdSRC**2*), we analyzed the cis-acting elements and domains in its CDS (Appendix A) [21]. A yeast two-hybrid analysis was used to determine whether MdTAC1a interacts with MdSRC2. The combinations of pGBKT7(BD)-53 + pGADT7(AD)-T and pGBKT7-Lam + pGADT7-T were used as positive and negative controls, respectively. All yeast colonies harboring different combinations of plasmids grew well on synthetic-shedding medium lacking leucine and tryptophan. Yeast colonies transformed with the plasmid combination, pGBKT7-MdTAC1a + pGADT7-MdSRC2, grew well on synthetic-shedding medium (SD-AHLT) in the absence of adenine, histidine, leucine, and tryptophan. The addition of X-α-gal gave results similar to the positive control (Figure 9A). Yeast cells transformed with the plasmid combination, pGBKT7-MdTAC1a-C+pGADT7, were not able to grow on SD-AHLT plates, similar to the negative control, which indicated that the C-terminus of MdTAC1a had no self-activity in yeast cells (Figure 9A). Taken together, these results indicate that the C-terminus of MdTAC1a binds to MdSRC2 in yeast.

Co-immunoprecipitation (Co-IP) experiments were then performed to provide further evidence for the interaction between MdTAC1a and MdSRC2 in vivo (Figure 9B). A bimolecular fluorescence complementation (BiFC) analysis was performed to further confirm the interaction between MdTAC1a and MdSRC2 in plant cells. pSPYNE and pSPYCE vectors, respectively, containing YFPN and YFPC, were used to construct MdTAC1a-cYFP and MdSRC2-nYFP, and vice versa. The fusion protein was transformed into *A. tumefaciens* and injected into tobacco leaves, and the fluorescence signal was observed under a scanning confocal microscope. When MdTAC1a-cYFP and MdSRC2-nYFP were transiently co-expressed, green fluorescence was observed in the nucleus, revealing the interaction of MdTAC1a and MdSRC2 in plants. No green fluorescence was detected when YFPN and YFPC were co-expressed in tobacco leaves (Figure 9C). Taken together, these results support the existence of an interaction between MdTAC1a and MdSRC2. In addition, the relative expression levels of the gene *MdSRC2* were detected in the parental and F_1_ population, The expression level of gene *MdSRC2* was significantly higher in trees with large branching angles(>45°) than in trees with small branching angles (≤45°) (Figure 10).

## 3. Discussion

### 3.1. Physiological Functions of MdTAC1 and MdTAC1-OE in Tobacco

In this study, we performed *MdTAC1a* and *MdTAC1b* subcellular localization experiments. Despite the 92.82% sequence similarity of nucleotide sequence between *MdTAC1a* and *MdTAC1b*, their subcellular localization was different. According to previous studies, *TaTAC1* is located in the cell membrane in wheat (*Triticum aestivum*) [18], but *SpsTAC2* is found in the nucleus in Salix [22]. These differences in the subcellular locations indicate that genes in the same family have different functions in different species.

We also used the *Agrobacterium*-mediated method to transform tobacco. *MdTAC1a/b*-OE-positive tobacco seedlings had significantly increased leaf angles. Overexpression of *TAC1* causing increased leaf angle is seen in herbaceous plants [15]; *the OsTAC1*-overexpressing transgenic rice line has been reported to exhibit a loose phenotype with a larger tiller angle [13]. Overexpression of *AtTAC1* in *Arabidopsis* can partially restore the phenotype of rice *Attac1* mutants [14]. In woody plants, *PpeTAC1*-overexpression in plum can induce horizontal growth of branches, and while silencing *PpeTAC1* in plum, the trees will grow toward the columnar trait as mutants of *PpeTAC1* showed in peach [18]. *MdTAC1a* shares a 84.84% similarity of deduced amino acid sequences with *MdTAC1b*, 72.61% with *PpeTAC1*, 12.18% with *OsTAC1*, 17.35% with *ZmTAC1*, and 8.12% with *AtTAC1.* High similarity of sequences of these *TAC1* genes in plants displays conserved biological function involving angle formation either in branches [14] and tillers [13] or leaves [15,23]. Unfortunately, no branching was observed in the *MdTAC1a/b*-OE-positive tobacco. Wild type Tobacco itself did not have branches under the same growth conditions as transgenic tobaccos (Appendix A). A possible reason was that *MdTAC1a/b* could not induce more branching but changed branch angles if lateral branches existed, like other plant *TAC1* genes. It is speculated that decapitation of *MdTAC1a/b*-OE and non-transformant plants may obtain plants with lateral branching to observe changes in branch angles.

### 3.2. Relationship of Growth and Tropic Responses in Transgenic Tobacco

Auxins can directly inhibit cytokinin biosynthesis by regulating the auxin resistant 1-dependent auxin-signaling pathway, thereby inhibiting the growth of axillary buds in Arabidopsis and apple [24,25]. In addition to cytokinins, strigolactones are auxin regulated and act as branching inhibitors. Inhibition of strigolactone biosynthesis or signal transduction can increase plant branching [26]. Strigolactone biosynthesis is affected by carotenoid-cleaving dioxygenase (CCD) [24].

By comparing the OE plants with WT tobacco, it was found that the genes on the gibberellin pathway were significantly different. Gibberellin is a diterpenoid phytohormone that plays an important role in plant growth [27]. The activity of gibberellin required the activation of gibberellin 20beta-dioxygenase (*GA20ox*), and gibberellin 3beta-dioxygenase (*GA3ox*). In addition, gibberellin 2beta-dioxygenase (*GA2ox*) was used to degrade the activity of gibberellins [28,29]. Gibberellin has been shown to promote negative gravitational growth of a weeping Japanese cherry (*Prunus spachiana*), i.e., inhibit angular enlargement [30,31].

Under either light or dark conditions, moreover, the upward-bending stem response of horizontally arranged OE plants was significantly different from that of the wild type (Figure 5). According to a previous report, *AtTAC1* responds to photosynthetic signals under dark or far-red light conditions by narrowing the branch angle [32]. Overexpression of *Md**TAC1a/b* displayed a more sensitive-to-light response by bending the stem with a larger curve in transformants compared to the dark condition. In addition, *IAA5* has been previously shown to be a good indicator of the gravity-induced auxin gradient in stems [33]. We speculate that *MdTAC1a* and *MdTAC1b* influence branch angle by acting on the transport of plant hormones.

### 3.3. Variation in the MdTAC1a Promoter among Cultivars and F_1_ Generation Plants

A promoter is the component of a gene that controls its expression, that is, the start time and degree of expression of transcription. A promoter does not control gene activity by itself, but instead functions in combination with transcription factors [34]. Prediction of specific sizes of different gene promoters is a difficult challenge. There are few reports on the relationship between promoter fragment size and function, but no reports on the relationship between promoters and branching angles. Multiple repeats of a promoter segment of *MYB 10* causes transcription factor autoregulation in red apples [34]. In the present work, we found that an InDel marker developed by specific primers based on promoter sequences of *TAC1a* co-segregated with large branch angles (>65°), either in analyzed apple cultivars, or in a F1 population. Promoters contain many *cis*-acting elements that may be useful for predicting the involvement of genes in processes, such as transcription control, light response, and plant hormone synthesis [20]. There were many differences in the *MdTAC1a* and *MdTAC1b* promoters of different apple cultivars. Compared to *MdTAC1a*, variation in *MdTAC1b* promoters was complicated in four-type apple cultivars (Figure 6). As for *MdTAC1a* promoters in various apple cultivars, several base deletions were identified in spur- and weeping-type cultivars; among them, the GAG motif was missing from a *cis*-acting, light-responsive element. The differential distribution of auxin after light exposure is the cause of phototropic movement in plants [35]. In above-ground stems, a high concentration of auxin promotes somatic cell growth, whereas a lower concentration inhibits it. In plants exposed to light, the lower part of the hypocotyl is shaded and accumulates a larger amount of auxin, which causes cells on the shaded side to elongate faster than those on the irradiated side, resulting in the phenomenon of stem bending [36,37]. In summary, the absence of a GAG motif in spur- and weeping-type cultivars probably influence these cultivars responding to light. 

### 3.4. The Effect of MdSRC2–MdTAC1a Interaction on Plant Branch Angle

The interaction between MdTAC1a and MdSRC2 has been verified by in vivo and in vitro experiments. SRC2 is a C2-domain protein in which C2 binds its own EF-hand motif as an intramolecular interaction [21]. The C2 domain independently forms a folded domain of 80–160 residues with characteristic binding Ca^2+^ and phospholipids [38]. In Arabidopsis, the cold-inducible protein AtSRC2 is a novel activator of the Ca^2 +^-dependent activation of AtRbohF(Respiratory burst oxidase homolog protein F) that enhances deliberate ROS(reactive oxygen species) production, which can increase the resistance of plants [39]. Ca^2+^ acts as a second messenger that converts the physical signal into physiological and biochemical signals and participates in gravity-signal transduction [40]. Such as Ca^2+^ concentration in chrysanthemum stems [41] and calreticulin and calmodulin mRNAs asymmetrically distributed in maize leaf [42]. In addition, the external application of calcium ion analogs or inhibitors can enhance or inhibit the gravitational response of plant hypocotyls [41,43,44,45]. In this experiment, both *MdTAC1a* and *MdSRC2* were highly expressed in apple trees with large branch angle (Figure 11), showing the interaction of *MdTAC1a* and *MdSRC2* involved in development of branch angle. Overexpression of *MdTAC1* could also increase plant gravitation, however, it remains to be confirmed whether the *MdSRC2* regulates the Ca^2+^ concentration and distribution to response gravity in apples.

## 4. Materials and Methods

### 4.1. Plant Materials

According to our observations and published apple (*Malus* x *domestica* Borkh.) tree statistics, we selected standard-type cultivars, ‘McIntosh’, ‘Summerland McIntosh’, and ‘Fuji’; columnar-type cultivars, ‘Wijcik’, ‘Waltz’, ‘Maypole’, and ‘Bolero’; weeping-type cultivar, ‘Granny Smith’; and spur-type cultivars, ‘Fukushima Spur’, ‘Miyazaki Spur’, and ‘Mutsu Spur’. We also included the following cultivars: ‘Zhongqiuwang’, ‘Red Star’, ‘Envy’, ‘Brilliant’, ‘Red Jade’, ‘Huashuo’, ‘Venus Golden’, ‘Huangyuxiang’, ‘Qincui’, ‘Tuoji’, ‘Dew’, ‘Max’, ‘Xiwang’, ‘Jinzhouhong’, ‘Daphne Red’, ‘Jinxianghong’, ‘Huajia’, ‘Huirui’, ‘Yanfu No. 10’, ‘Yanfu No. 6’, and ‘Yanfu No. 8’. Furthermore, we developed an F_1_ segregation population from four-year-old seedlings of a cross between ‘Jin Lei No. 1’ (female parent, narrow-tree architecture) and ‘Granny Smith’ (male parent, broad-tree architecture). The branching angle sizes of F1 progeny were measured and grouped in two subgroups: narrow branch angle (≤65°) and broad branch angle (>65°).

All cultivars and F_1_ progeny were grown at the experimental station of China Agricultural University, Beijing, China (latitude 40.138044° N; longitude 116.185320° E). In May 2019, leaves were harvested from one-year-old woody shoots, immediately frozen in liquid nitrogen, and stored at −80 °C. All branch-angle statistics are on one-year-old branches, and the height is 50–120 cm. In vitro cultured wild-type tobacco (*N. *benthamiana***)* plants were used for genetic transformation and subcellular localization experiments. Tobacco was grown at 22 °C with 16 h of light and 8 h of darkness. The leaves were used to detect changes in hormone expression levels of transgenic tobacco for 10 weeks after transgenesis.

### 4.2. DNA and RNA Extraction

Genomic DNA (gDNA) of apple was extracted by the cetyltrimethylammonium bromide (CTAB) method as described by Doyle [46], and total RNA was extracted using a modification of this method [47]. RNA degradation and contamination was monitored on 1% agarose gels. RNA purity and concentration were checked using a NanoPhotometer spectrophotometer (Implen, Westlake Village, CA, USA).

### 4.3. Real-Time Quantitative PCR (qRT-PCR) Analysis

cDNA of apple and tobacco were transcribed from 2 µg of total RNA in 20 µL reaction mixtures using a TransScript Uni All-in-One First-Strand cDNA Synthesis SuperMix for qPCR (One-Step gDNA Removal) (TransGen, Beijing, China). qRT-PCR analyses were carried out using a *PerfectStart* Uni RT&qPCR kit (Perfect Real Time, TransGen) with the qRT-PCR primers shown in Appendix A on an Applied Biosystems One-Step Plus instrument (Applied Biosystems, Foster City, CA, USA). The cycling conditions were as follows: 94 °C for 30 s, followed by 40 cycles of 95 °C for 5 s, 60 °C for 15 s, and 72 °C for 10 s. Gene transcript levels were normalized to that of *MdActin*. Each analysis was repeated three times. The 2^−ΔΔCt^ method was used to calculate relative transcript levels of each gene [48]. Three replicates of samples from different plants under the same conditions were collected for qRT-PCR analyses.

### 4.4. Generation of MdTAC1a/b–GFP Fusion Constructs, Transformation of Tobacco, and Subcellular Localization of MdTAC1a/b Protein

Comparisons of the coding sequences (CDSs) of *MdTAC1a/b* from the different plant materials revealed numerous SNPs. The full-length CDSs of *MdTAC1a/b* were amplified from these materials using specific primers (Appendix A). The CDSs of *MdTAC1a/b* were then inserted between BamHI and SalI restriction sites upstream of the GFP gene in the binary vector pCambia1300. Each resulting construct was introduced into *Agrobacterium tumefaciens* strain EHA105 using the freeze–thaw method [49,50]. The leaf disc method was used to transform wild-type tobacco, and the regenerated plants were screened according to their kanamycin resistance. Part of the fusion vector was also transformed into *Agrobacterium* GV3101 competent cells, and the suspension containing the fusion vector was instantaneously injected into the leaves of tobacco. After the transformation, the tobacco leaves were kept at 25 °C in the dark for 48 h. Following exposure to light for 24 h, the tobacco leaves were inspected under an FV1000 confocal microscope (Olympus, Tokyo, Japan).

### 4.5. InDel Marker Development and Genetic Correlation Analysis

The 2000-bp region upstream sequences of the start codon ATG of *MdTAC1a/b* located on chromosomes 7 and 1, respectively, of the apple genome [20] were downloaded from GenBank (https://www.ncbi.nlm.nih.gov/genome/?term=apple (accessed on 5 September 2019)) and used to design amplification primers (Appendix A). The *MdTAC1a* and *MdTAC1b* promoter sequences of the 11 apple cultivars were amplified using these primers and high-fidelity enzymes, and the amplified products were purified and ligated to a cloning vector for sequencing. The generated promoter sequences were compared among cultivars, and SSR primers were subsequently designed to amplify the 400 bp near the promoter deletion region. SDS-PAGE was used in conjunction with these primers to distinguish between tree types with different branch angles and the F_1_ generation. The relationship between tree phenotype and genotype was then used to calculate the probability of genetic linkage.

### 4.6. Yeast Two-Hybrid Assay (Y_2_H), Bimolecular Fluorescence Complementation (BiFC) Analysis, and Co-Immunoprecipitation (Co-IP)

For yeast two-hybrid screening, BD-MdTAC1a was amplified by PCR using four primers (Appendix A). After purification, the amplified PCR product was ligated with the empty pGBKT7 plasmid, which had been double digested with EcoRI and BamHI restriction enzymes. The correctly sequenced pGBKT7-MdTAC1a plasmid was used as a bait to screen the interaction library of apple in the pGADT7 vector. Transformation of the Y_2_H strain was performed with a matchmaker gold yeast two-hybrid kit (Takara, Tokyo, Japan), and putative positive clones were obtained and sequenced. Homologous genes in the generated sequences were identified by BLAST searching.

PCR products generated using primers BiFC-MdTAC1a-C-F/BiFC-MdTAC1a-C-R and BiFC-MdSRC2-N-F/BiFC-MdSRC2-N-R (Appendix A) were connected to pSPYNE and pSPYCE vectors digested with XbaI and XhoI to generate MdTAC1a-pSPYCE and MdSRC2-pSPYNE, respectively. For transient transformation, the ligated vectors were injected into five-week-old tobacco leaves via *A. tumefaciens* strain GV3101/p19. After the transformation, the tobacco leaves were kept at 25 °C in the dark for 48 h. Following exposure to light for 24 h, the tobacco leaves were inspected under an FV1000 confocal microscope (Olympus).

For the Co-IP assay, the CDSs of *MdTAC1a* and *MdSRC2* were amplified and individually inserted into the vector pCAMBIA1300-221 with HA and FLAG tags using the primer pairs listed in Appendix A. These plasmids were transformed into *A. tumefaciens* GV3101/p19 and co-infiltrated into tobacco as described above. After 72 h, total protein was extracted using a plant total protein extraction kit (Hua Xing Bo Chuang Bio, Beijing, China) and incubated with Anti-BIFC immunomagnetic beads (Bimake Bio, Beijing, China) at 4 °C overnight. The beads were washed three times with PBST, and the precipitated proteins were further analyzed by western blotting using anti-HA monoclonal antibody.

### 4.7. Statistical Analysis

All samples were analyzed in triplicate, with data expressed as means ± standard errors unless otherwise noted. Statistical significance was assessed by analysis of variance using SPSS 17 (SPSS, Chicago, IL, USA). Differences were considered significant at *p* < 0.05 and extremely significant at *p* < 0.01.

## 5. Conclusions

In this study, we analyzed the biological functions of the genes *MdTAC1a*/*b*, including their subcellular locations and heterologous overexpression. The results revealed changes in phenotype of transgenic plants, key genes related GA and photosynthesis signal, and a differential response to gravity under light and dark conditions in transgenic and WT plants. We also identified *MdTAC1a*/*b* genes in apple cultivars, examined their tissue-specific expression patterns, and verified variations in their promoter sequences among apple cultivars and F_1_ hybrid. Finally, we revealed the interaction between MdSRC2 and MdTAC1a proteins through a yeast two-hybrid assay and verified this interaction in BIFC and Co-IP analyses. According to these results, *MdTAC1a*/*b* genes may play an important role in branch-angle development in apple. The molecular marker developed in this study should be useful for early selection of apple-branch angles in offsprings and thus, contributes to manipulate tree architecture of apples. 

## Figures and Tables

**Figure 1 ijms-23-01870-f001:**
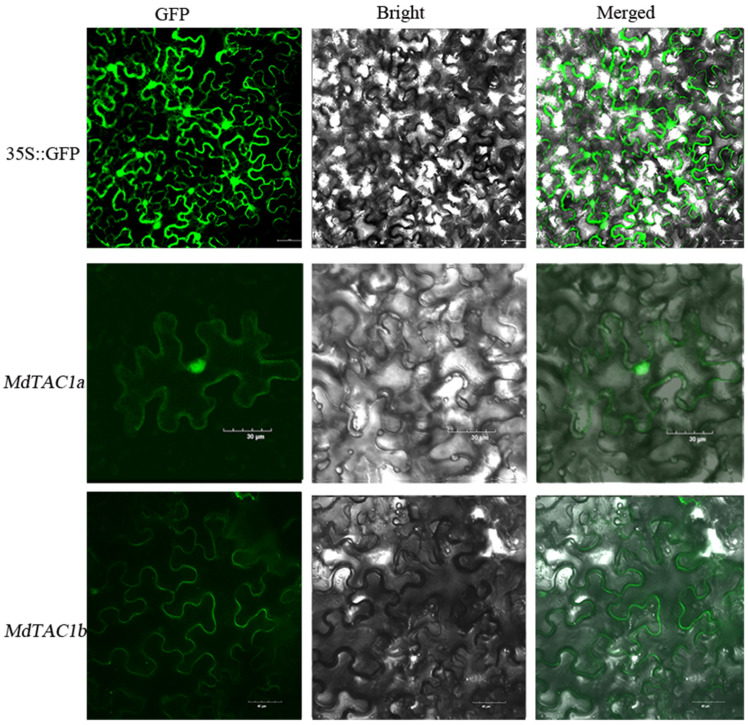
Subcellular localization of *MdTAC1a* and *MdTAC1b* in tobacco leaves. 35S::GFP represented the pCambia1300 with GFP vector and the pCambia1300 empty vector was detected in the nucleus and membrane. *MdTAC1a* was detected on the nucleus and membrane, and *MdTAC1b* was detected on the membrane.

**Figure 2 ijms-23-01870-f002:**
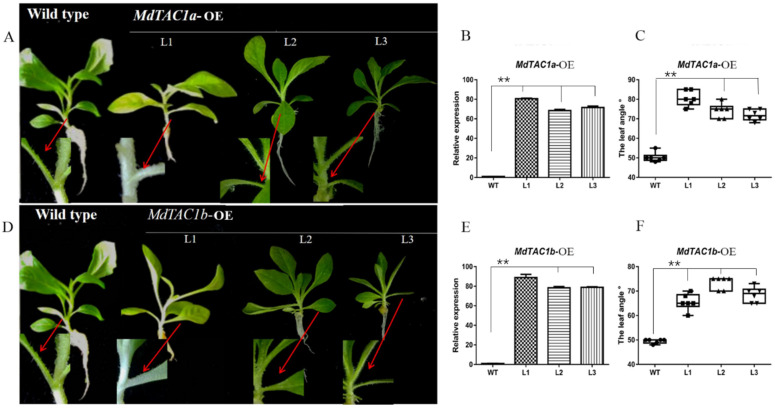
Phenotypes, leaf angles, and relative expression levels of *MdTAC1a* and *MdTAC1**b* in transgenic tobacco plants. (**A**,**D**) Phenotypes of *MdTAC1a*-OE and *MdTAC1b*-OE transgenic plants. (**B**,**E**) The relative expression levels of *MdTAC1a* in *MdTAC1a*-OE and *MdTAC1b* in *MdTAC1b*-OE transgenic plants. (**C**,**F**) Leaf angles of *MdTAC1a*-OE and *MdTAC1b*-OE transgenic plants. WT: the wild type; L1–L3: the transgenic lines. The significance of difference was analyzed with two-tailed *t*-test (** *p* < 0.01).

**Figure 3 ijms-23-01870-f003:**
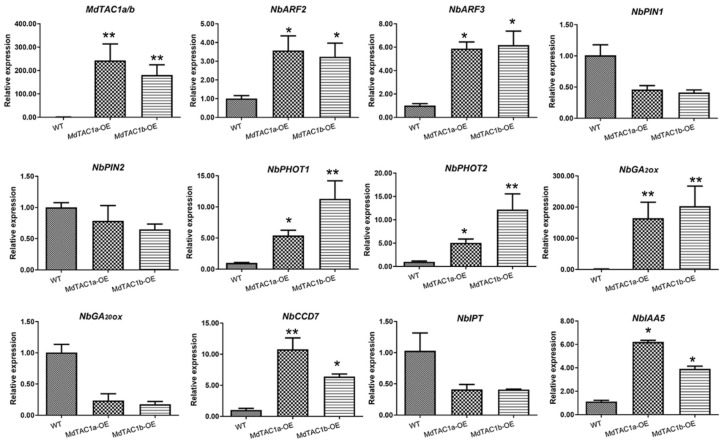
Expression profiles of 12 key genes involved in synthesis and signaling pathways in the WT and transgenic plants eight weeks after transgenic. Error bars represented the standard error of the mean for three biological replicates. The results showed the gene relative expressions in the *MdTAC1a* and *MdTAC1b* transgenic lines were significantly more than that of the wild type in tobacco. The significance of difference was analyzed with two-tailed *t*-test (* 0.01 < *p* < 0.05, ** *p* < 0.01). Data were represented as average values with SD.

**Figure 4 ijms-23-01870-f004:**
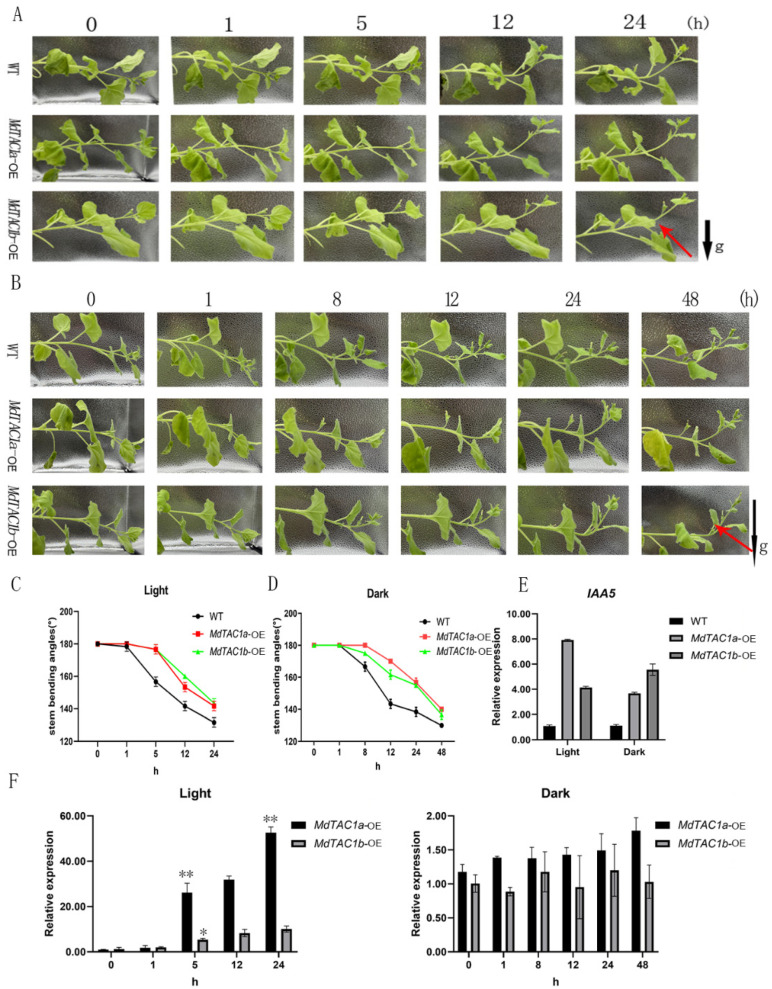
Gravitropic and light responses in wild-type (WT), *MdTAC1a*-OE, and *MdTAC1b*-OE tobaccos. (**A**,**B**) Tobacco responded to gravity for 24 h under light and 48 h for darkness. (**C**,**D**) The angle in the stem was bent at the position of the red arrow in (**A**,**B**). Bars indicated the means ± SDs (*n* = 3). (**C**) The statistics of the degree of bending of plant stems placed horizontally under light at 0, 1, 5, 12, and 24 h. (**D**) The statistics of the degree of bending of plant stems placed horizontally at 0, 1, 8, 12, 24, and 48 h in the dark. The rate of stem bending was faster in WT than in *MdTAC1a*-OE and *MdTAC1b*-OE plants, regardless of light or dark treatments. (**E**) The relative expression level of IAA5 in the stem at the position of the red arrow in (**A**,**B**). (**F**) Dynamic changes in relative expression levels of *MdTAC1a* and *MdTAC1b* in OE plants during stem-bending period under light and dark condition. The results showed the gene relative expression in the *MdTAC1a* and *MdTAC1b* transgenic lines was significantly more than that of the wild type in tobacco. The significance of difference was analyzed with two-tailed *t*-test (* 0.01 < *p* < 0.05, ** *p* < 0.01). Data were represented as average values with SD.

**Figure 5 ijms-23-01870-f005:**
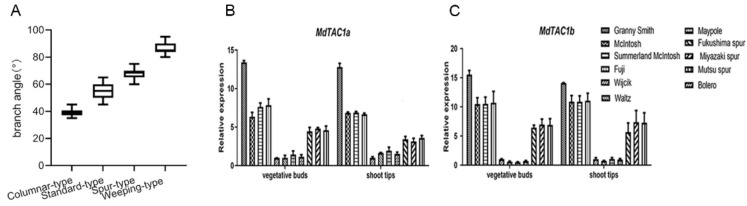
Statistics of branching angles of four tree typesand the expression of *MdTAC1a* and *MdTAC1b* in four apple cultivars. (**A**) Statistics of branching angles of four tree types (**B**,**C**) The relative expression patterns of MdTAC1a/b in vegetative organs of apple cultivars.

**Figure 6 ijms-23-01870-f006:**
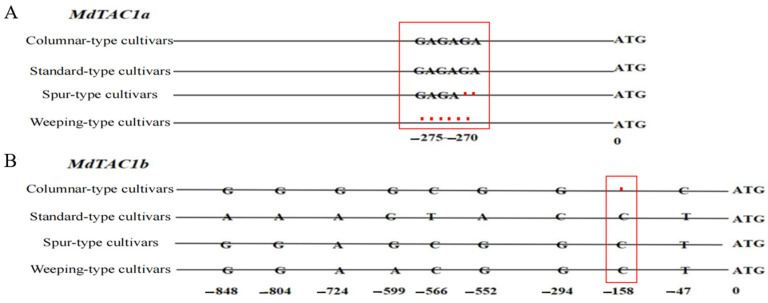
The Differences of *MdTAC1a* and *MdTAC1b* promoters among four apple cultivars. (**A**) The sequence of InDel in promoter regions of *MdTAC1a* in four-type cultivars. The initiation codon (ATG) A was defined as 0. The red box was a representation of the InDel area (**B**). The sequence SNPs in promoter regions of MdTAC1b in four-type cultivars.

**Figure 7 ijms-23-01870-f007:**
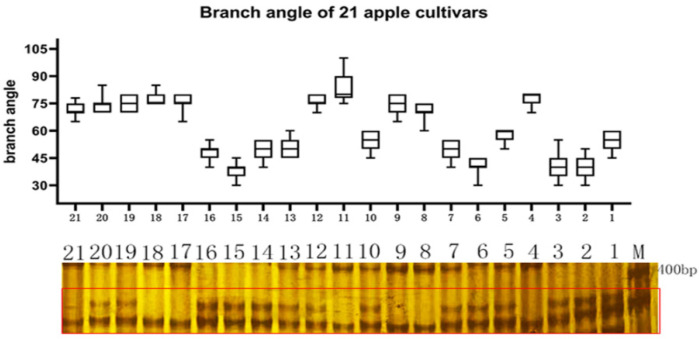
Genotyping of *TAC1a* in apple cultivars by an InDel markerThe upper imageshowedthe branch angle size of 21 apple cultivars, and the lower electrophoresis pattern showed the separation of *TAC1a* marker in the 21 cultivars, 1-21 represented sequentially ‘Zhongqiuwang’, ‘Red star’, ‘Envy’, ‘Brilliant’, ‘Red jade’, ‘Huashuo’, ‘Venus golden’, ‘Huangyuxiang’, ‘Qincui’, ‘Tuoji’, ‘Dew’, ‘Xiwang’, ‘Jinzhouhong’, ‘Daphne red’, ‘Jinxianghong’, ‘Huirui’, ‘Max’, ‘Huajia’, ‘Yanfu NO.6’, ‘Yanfu NO.10’, ‘Yanfu NO.8’.

**Figure 8 ijms-23-01870-f008:**
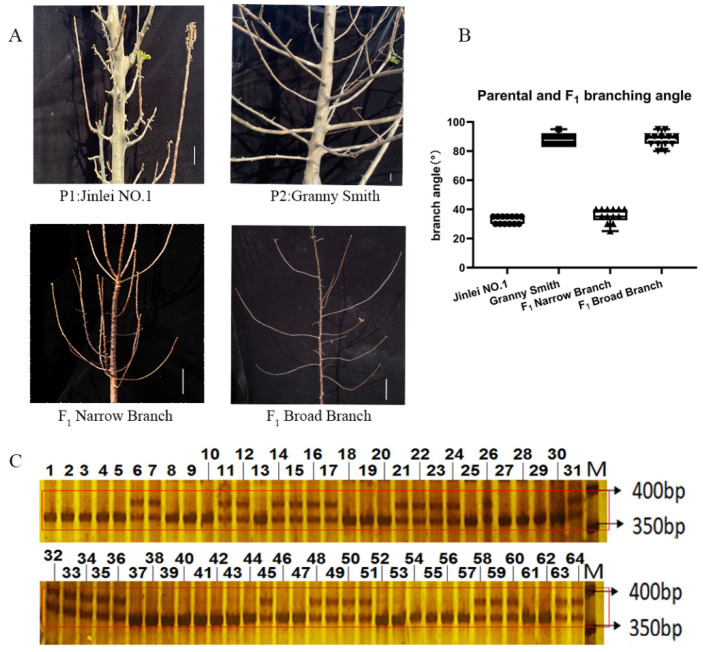
Genotyping of *TAC1a* in apple parental and F_1_ individuals by an InDel marker. (**A**) Apple parental and F_1_ individuals. (**B**) was the branch angle statistics of the parents and F_1_. (**C**) was the separation of *TAC1a* marker in the parental and F1 progeny. M represented DNA ladder marker; Line 1 to 29 and line 32 to 64 represented F_1_ generation; 30 represented parental Granny Smith; 31 represented parental Jinlei No. 1 The parent and F_1_ were separated at 400 bp of the red box in Figure (**C**).

**Figure 9 ijms-23-01870-f009:**
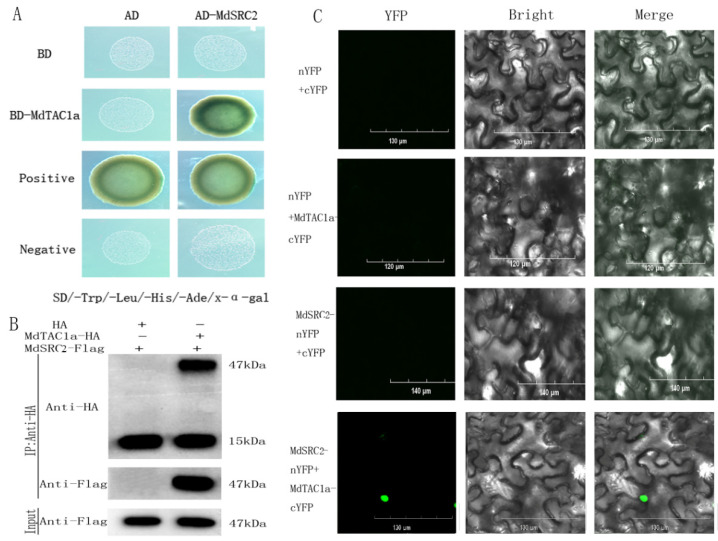
MdTAC1a interacted with MdSRC2. (**A**) Yeast two-hybrid analysis showing the interaction between MdTAC1a and MdSRC_2_. pGBKT7-Lam + pGADT7-T were used as negative controls. pGBKT7-53 + pGADT7-T were used as positive controls. (**B**) Co-IP analysis in tobacco showed the interaction between MdTAC1a and MdSRC_2_. Empty vectors were used as negative controls. (**C**) BiFC analysis in tobacco epidermal cells showed the interaction between MdTAC1a and MdSRC_2_. Empty vectors were used as negative controls.

**Figure 10 ijms-23-01870-f010:**
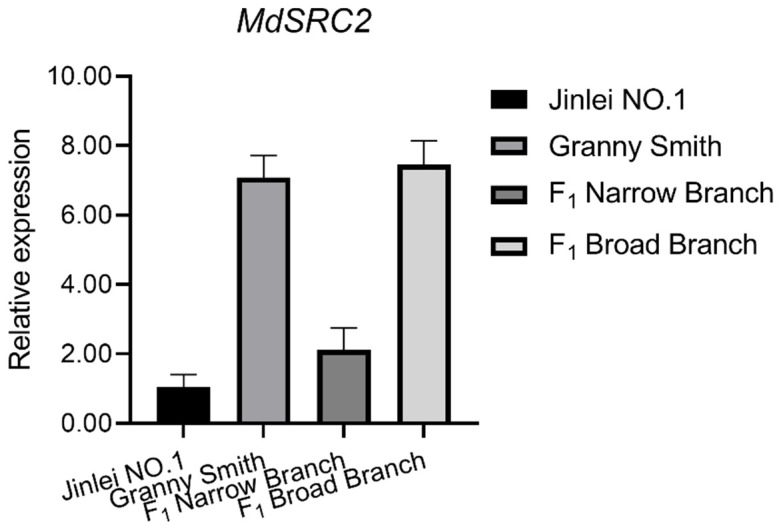
Expression of *MdSRC2* in parental cultivars and F_1_ progeny.

**Figure 11 ijms-23-01870-f011:**
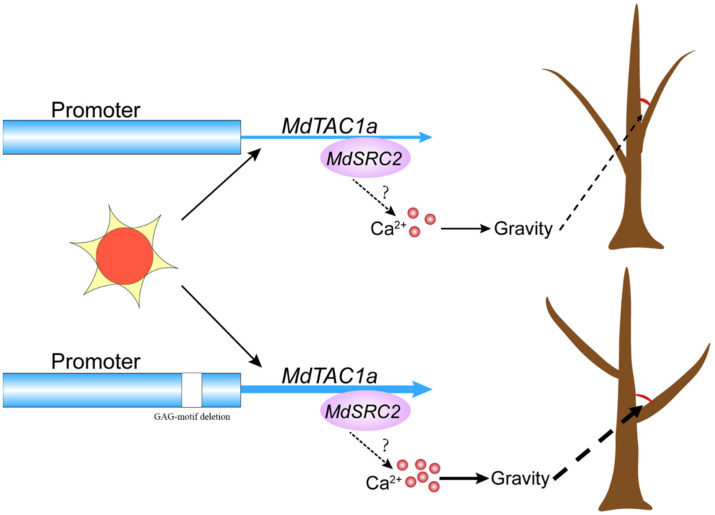
The model of *MdTAC1a* in regulating the branching angle of apple trees. In apple MdTAC1a interacts with MdSRC2 to affect intracellular Ca^2+^ concentration remains to be proven.

## Data Availability

Primers and accession numbers of genes analyzed in this study are listed in Appendix A. Seeds used in transgenic lines and all other data used in this study are available from the corresponding authors upon request.

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
