# Peer review of "Functional Characterization of *MdTAC1a* Gene Related to Branch Angle in Apple (*Malus* x *domestica* Borkh.)"

_ijms, 2022, doi:10.3390/ijms23031870_

Round 1

Reviewer 1 Report

The present research author performed a complementation study for MdTAC1a/b  gene in a heterologous system (Tobacco) further subcellular localization experiments were carried out and changes in the branch angle of the transgenic OE lines was confirmed thus the gene functions of MdTAC1a/b were determined. In addition, the author developed new molecular markers for branch angle and used yeast two-hybrid technology to study the MdTAC1a-interacting protein MdSRC2. The topic is relevant and the study conducted may be useful for the apple research.

Firstly, method, results, and discussion is not written in series the way experiment perform it must need to be revised properly,

Second,  I have some major concerns about this study:

  1. The article starts with the MdTAC1a subcellular localization, however, it should be explained properly how this is identified and further decided to characterize with the complementation study in the heterologous system. The author does not mention how the MdTAC1a/b genes are identified.
  2. As the author said they did not observe any change in the different cultivars MdTAC1a/b CDS therefore is it appropriate to say that MdTAC1a/b  functional cause for branch angle difference in the apple?
  3. Secondly, the author used just MdTAC1a/b gene coding region for creating the vector without the promoter region then based on the relative expression difference its stated that MdTAC1a/b  gene is responsible for the branch angle difference is it justified?
  4. Is there any similar studies researcher have reported that due to variation in the promoter region they observed the ideotype difference for the particular trait?
  5. Branch angle is a quantitative trait and research over the past several decades has identified QTLs that regulate branch angle in different plant species is it not possible there is any major QTL present in that region that is responsible for branch angle difference in apple different cultivars?
  6. The author hypothesizes is that that the ectopic overexpression of the apple MdTAC1a/b gene was responsible for this increase in tobacco leaf angle for several reasons. First, the MdTAC1a/b gene of apple may also have a role in regulating the branch angle of plants and, second, an OsTAC1-overexpressing transgenic rice line has been reported to exhibit a loose phenotype with a larger tiller angle and AtTAC1 in Arabidopsis did author check sequence similarity between apple and mentioned genes or other plant genes. Secondly, please explain how the MdTAC1a/b- OE-positive tobacco increased leaf angles will correlate with branch angle?

There are several errors in writing hence before it considers for publication in IJMS it must go through substantial revision. Hence I would like to endorse for Major revision.

Abstract:

An abstract is very important and most of the readers follow abstract, hence I would suggest to please write your abstract in a more standard manner. An abstract normally has 1-3 introductory sentences that discuss the field or general problem. Then 2-4 sentences of your major findings, then perhaps a concluding sentence. At the moment, your abstract has no introduction and has too many technical details for the general reader to appreciate.

Please write the Apple scientific name once at first appearance in the abstract then follow the common name.

Introduction:

“Apple (Malus domestica Borkh.) is widely cultivated, and this species ranks third with respect to global production” please provide the reference.

Page2: As the author mentioned the botanical or scientific name of some species (apple and tobacco), please write a scientific name for rice and other plant species mentioned in the introduction, check this throughout the MS, and maintain consistency or uniformity.

“In a previous investigation, we separately cloned the TAC1a/b genes of apple. No cDNA differences were detected among analyzed ideotypes, but variation was observed in the promoters” please provide the reference in which study this work has been carried out.

Results:

This sentence is redundant please remove “The relative expression level of MdTAC1b in L1 was almost 80 times that of the WT (Fig. 2)”.

“Figure 2. Phenotypes, leaf angles, and relative expression levels of MdTAC1a and MdTAC1b in transgenic Arabidopsis plants”.  Please check this legend author mentioned transgenic tobacco and in the figure legends it's written “Arabidopsis plants”.   

Secondly, the significance level is not mentioned in the graph please explain why? Also, check all the graphs and mention the significance level on the bar graph as well as in the figure legends.

Each figure has a different font size and style please follow the uniform font and style throughout the Figures.

Figure 3, side heading written as “MdTAC1a-oe,” MdTAC1b-oe” in the legends its mentioned as “MdTAC1a-OE, and MdTAC1b-OE” why the small “oe” is used in the side heading?

This is method part need to explain what results author observed or obtained and not what method followed ”In the present study, we analyzed the expressions of MdTAC1a and MdTAC1b in 11 apple varieties with different branch angles, namely, three standard type cultivars (‘McIntosh’, ‘Summerland McIntosh’, and ‘Fuji’), four columnar-type cultivars (‘Wijcik’, ‘Waltz’, ‘Maypole’, and ‘Bolero’), three spur-type cultivars (‘Fukushima spur’, ‘Miyazaki spur’, and ‘Mutsu spur’), and weeping-type ‘Granny Smith’. RNA was extracted from shoot tips and vegetative buds of these 11 apple varieties and reverse transcribed”.

Figure 6. Please subscript 1 in  “F1” write as “F1

Figure 6. F and H separation on which (PAGE?) need to mention and highlight the band of distinguishing the cultivars,

This is again method part “DNA was extracted from the different cultivars and the F1 generation and PCR amplified with the InDel primers. Among the 21 cultivars, PCR amplification yielded the complete 370 bp in 14 materials, whereas a 6-bp deletion was found in 7 cultivars (Fig. 6F).” please remove and just explain the results. Throughout the results, section checks carefully remove the method part and write your results properly.

At some places, the author used the term cultivars at some places varieties there is a huge difference between cultivars and varieties please appropriate term.

Discussion:

This section needs to revise it's not explained properly according to the experiment performed, author needs to discuss all the results found in this study.

As the author wrote the common name and scientific names for plant species previously please write common name also for “Triticum aestivum and  Salix psammophila”

Figure 8, “MdSRC2” “2” is subscripted while writing in the text it is not pleasing maintain uniformity.

Materials and Methods:

Please write “The 2,000-bp region upstream sequences of the ATG” as The 2,000-bp region upstream sequences of the start codon ATG……”

Experimental conditions not mentioned in detail (photoperiod, soil condition, day length, etc)

Which tobacco ecotype was used for the transformation?

Author Response

Response to Reviewer 1 Comments

Point 1: The article starts with the MdTAC1a subcellular localization, however, it should be explained properly how this is identified and further decided to characterize with the complementation study in the heterologous system. The author does not mention how the MdTAC1a/b genes are identified.

Response 1: Thanks for reminder!

In fact, two homologs of TAC1 in apple have been idientifed in previous study of our laboratory. This paper has been quoted in revision of this manusript, please see line73.  However, the function of two genes has not been explored in previous work, so experiments in this study will proceed directly from the functional analysis.

Point 2:As the author said they did not observe any change in the different cultivars MdTAC1a/b CDS therefore is it appropriate to say that MdTAC1a/b  functional cause for branch angle difference in the apple?

Response 2: We thank the Reviewer for the good question.

The full length of a gene includes promoter, intron, and exons. Actually, there were no difference observed in CDS among apple cultivars representing four ideotype apple trees (Wang et al., 2018) , but the promoter of MdTAC1 in four-type apple trees with different branching angles did have InDel for MdTAC1a and SNP for MdTAC1b, which means the promoter rahter than CDS of MdTAC1 affects brach angles in apple. Therefore, considering broad sense of a gene, we changed the title of this article to ‘Functional characterization of MdTAC1 gene related to branch angle in apple (Malus X domestica Borkh.)’, also sugessted by one Reviewer.

Point 3:Secondly, the author used just MdTAC1a/b gene coding region for creating the vector without the promoter region then based on the relative expression difference its stated that MdTAC1a/b gene is responsible for the branch angle difference is it justified?

Response 3: Thanks for the question. It points out an efficient way to analyze bioliogical function of a given gene through transformation of its CDS driven by its native promoter. However, MdTAC is funcitonal unknown, we thought, it would be better to start with the CDS for functional analysis, and then promoter. Transgenic tobcaccos displayed enlarged leaf angles by overexpression of the CDS of MdTAC1, indicating MdTAC1 involved in plant angle formation. Apple cultivars and F1 progeny with different branching angleswere used to verify the deletion of GAG- motif in promotor of MdTAC1a affecting variation of banch angles in apple.

Point 4: Is there any similar studies researcher have reported that due to variation in the promoter region they observed the ideotype difference for the particular trait?

Response 4:  According to our knowledge, TAC (GenBank MG837476/MG837477.) in apple is first reported by our previuos work(Wang et al., 2018). Other TAC1 genes in wooden species, such as peach (Dardick et al, 2013) and poplar (Xu et al., 2017), exert different mechanisim underlying regulating branch angles. In apple, variation in the promoter region associated with particular trait is not exception. Multiple repeats of a promoter segment of MYB 10 causes transcription factor autoregulation in red apples (Espley et al., 2009). In present work, we found that an InDel marker developed by specific primers based on promoter sequences of TAC1a cosegregated with large branch angles (>65°). This question has been discussed in rivised version, please see lines 441-445.

Xu D , Qi X , Li J , et al. PzTAC and PzLAZY from a narrow-crown poplar contribute to regulation of branch angles.[J]. Plant Physiology & Biochemistry, 2017, 118,571.

Espley RV, Brendolise C, Chagné D, Kutty-Amma S, Green S, Volz R, Putterill J, Schouten HJ, Gardiner SE, Hellens RP, Allan AC. Multiple repeats of a promoter segment causes transcription factor autoregulation in red apples. The Plant Cell, 2009,21, 168-183

Point 5:Branch angle is a quantitative trait and research over the past several decades has identified QTLs that regulate branch angle in different plant species. Is it not possible there is any major QTL present in that region that is responsible for branch angle difference in apple different cultivars?

Response 5: At present, QTLs conresponding to weeping traits in apple have been found by pooled genome sequencing from the cross ‘Cheal’s Weeping’ × ‘Evereste’, a major locus, Weeping (W), on chromosome 13, and others on chromosomes 10 (W2), 16 (W3), and 5 (W4) ( Dougherty et al., 2018)., which . MdTAC1 is located on chromosomes 7 (MdTAC1a) and 1 (MdTAC1b), which is not inclusive in the predicted genes in Weeping loci in apple.

Dougherty L, Singh R, Susan B, Dardick C, Xu K. Exploring DNA variant segregation types in pooled genome sequencing enables effective mapping of weeping trait in Malus. Journal of Experimental Botany, 2018, 69 (7): 1499–1516

Point 6: The author hypothesizes is that the ectopic overexpression of the apple MdTAC1a/b gene was responsible for this increase in tobacco leaf angle for several reasons. First, the MdTAC1a/b gene of apple may also have a role in regulating the branch angle of plants and, second, an OsTAC1-overexpressing transgenic rice line has been reported to exhibit a loose phenotype with a larger tiller angle and AtTAC1 in Arabidopsis. Did author check sequence similarity between apple and mentioned genes or other plant genes. Secondly, please explain how the MdTAC1a/b-OE-positive tobacco increased leaf angles will correlate with branch angle?

Response 6: We agree to the comments. The hypothesis made in this manuscript and reasons listed fail to convey the idea. Thus, we rewrote this part, please see lines 374-391.

Overexpression of MdTAC1a/b resulted in enlarged leaf angles in transgenic tobaccos. This phenotype in the MdTAC1a/b- OE-positive tobacco resambles OsTAC1-overexpressing transgenic in rice and AtTAC1 in Arabidopsis. According to the phylogenetic analysis, MdTAC1a/b are cluster  with PpeTAC1, OsTAC1, and ZmTAC1 (Wang et al., 2018). MdTAC1a shares 92.82% similarity of deduced amino acid sequenses with MdTAC1b, 81.22% with PpeTAC1, 38.58% with OsTAC1, 38.56% with ZmTAC1, and 25.51% with AtTAC1. High similarity of seqences of these TAC1 genes in plants displays conserved biological founction involving in angle formation either in branchs (Dardick et al.,2013) and tillers(Yu et al.,2010), or leaves (Lixia et al., 2011). Unfortunately, no branching was observed in the MdTAC1a/b-OE-positive tobacco. Wild type T. (N. benthamiana) itself did not have branches in same growth conditions as transgenic tobaccos (Fig.S1) . The possible reason was that MdTAC1a/b could not induce more branching but change branch angles if lateral branches existed, like other plant TAC1 genes. It is speculated that decapitation of and MdTAC1a/b- OE and non-transformant plants may obtain plants with lateral branching to observe changes in banch angles.

Dardick, Callahan, Horn, et al. PpeTAC1 promotes the horizontal growth of branches in peach trees and is a member of a functionally conserved gene family found in diverse plants species. PLANT J, 2013, 2013,75(4)(-):618-630.

 Yu B , Lin Z , Li H , et al. TAC1, a major quantitative trait locus controlling tiller angle in rice[J]. The Plant Journal, 2010, 52,891-898.

Lixia K , Wei X , Zhang S , et al. Cloning and Characterization of a Putative TAC1 Ortholog Associated with Leaf Angle in Maize (Zea mays L.). Plos One, 2011, 6,e20621.

In addition, all suggestions for writing proposed by the reviewer have been revised in new vesion of the manuscript.

Reviewer 2 Report

Dear the editor,

The manuscript entitled “ Characterization of MdTAC1a, related to branch angle in apple (Malus domestica Borkh.)” describes the molecular characters and possible roles of the MdTAC1a/b genes in relation to regulation in branch angles.

One of most interest data is correlation between the promoter sequence of the MdTAC1a and plant branching types. First, I could not simply understand the segregation rate in F1 progeny. Is one of the parent cultivar (Jinlei No.1 or Granny Smith) heterozygous for the promoter sequence of the MdTAC1a? In the manuscript, heterozygosity of the parent lines is not mentioned.

Second, correlation between the expression levels of MdTAC1a/b and plant branching types is also impressive. However, it seems not logical that the only differences in the MdTAC1a promoter sequence are associated with the branching types. The authors should show the data on the MdTAC1b promoter sequence in the F1 progeny.

In discussion section, the absence of the GAG motif is discussed with light response. However, any experimental evidence on relationship between the promoter sequences and light response is provided in the manuscript. The light condition indeed affect the MdTAC1a expression?

Physical interaction between MdTAC1a and MdSRC2 seems informative. In the results of MdSRC2 experiments, it is necessary to provide more information for characterization of MdSRC2 protein since a number of C2 domain containing proteins exist in plants. “MdSRC2” came as bolt out of the blue.

In page 9, the first paragraph, the last sentence, “these results indicate that the C-terminus of MdTAC1a binds to MdSRC2” is not explained by the provided data. What experiment indicates the importance of the C-terminus in terms of binding to MdSRC2?

Moreover, it is not tested in the manuscript at all that interaction between MdTAC1a and MdSRC2 is involved in regulation of branching angles. The last sentence in Abstract should be reconsidered.

Cellular localization of MdTAC1a and MdTAC1b is observed in tobacco. I think more careful interpretation to the observation is required. Especially GFP signals outside of the nucleus seems not only on the plasma membrane but in the cytosol. To exclude the possibility of cytosol localization of the GFP-fusion MdTAC1a/b, the authors may need to add some control experiments. In the provided data, “Control” seems to express free GFP. The free GFP can be accumulated in nucleus and cytosol.

The legend in figure 1 indicates that “Control” is empty vector pCambia1300, but “Control” obviously showed GFP signal suggesting “Control” has 35S::GFP.

In BiFC experiment, YFP signals seems to be detected outside of the nucleus.

Overexpression of MdTAC1a or MdTAC1b in tobacco provides nice results. Phenotypic differences between MdTAC1a OE plants and MdTAC1b OE plants are interesting, In the OE-plants, is the expression levels of MdTAC1a or MdTAC1b changed in light or dark condition? In other words, do the authors explain phenotypic differences between MdTAC1a OE plants and MdTAC1b OE plants as expression changes in light or dark condition even when the transgenes are under control of the 35S promoter?

In page3, comparison in expression levels between the transgenic plants and WT plants seems weird since WT plants do not contain the transgene (MdTAC1a or MdTAC1b).

Minor points

In figure2, Arabidopsis should be tobacco.

In Figure 6F and 6H, it is difficult to see which band represents non-deleted sequence and 6-bp deleted sequence.

In page 11, the last sentence, “GA” should be Ca.

Author Response

Response to Reviewer 2 Comments

Point 1: One of most interest data is correlation between the promoter sequence of the MdTAC1a and plant branching types. First, I could not simply understand the segregation rate in F1 progeny. Is one of the parent cultivar (Jinlei No.1 or Granny Smith) heterozygous for the promoter sequence of the MdTAC1a? In the manuscript, heterozygosity of the parent lines is not mentioned.

Response 1: Yes. The F1 population was obtained by the cross of ‘Jinlei No.1’(cloumnar tpye) × ’Granny Smith’ (Weeping type). The branch angle was narrow (≤45°) in ‘Jinlei No.1 and larger (>75°) in ‘Granny Smith’(please see Material and Method). There is no InDel in the promoter region of MdTAC1a in columnar cultivars and a deletion of 6 bp in MdTAC1 promoter in weeping cultivar ‘Granny Smith ‘ (Fig. ). The InDel marker was developed by specific primers to MdTAC1 promoters and tested for the polymorphism in parents and in F1 population by SDS-PAGE ( Fig.8). The branching angle sizes of F1 progeny were measured and grouped in two subgroup: narrow branch angle(≤65°) and broad branch angle(>65°)( Material and Method). This marker, a product of 300 bp by PCR amplification, was consegregated 100% with large branch angle trees, and **% with middle-type trees. No segement of 300 bp was detected in small branch angle trees (Fig). The results showed that the InDel marker was efficient to distinguish the branch angle size in F1 progeny.

Point 2: Second, correlation between the expression levels of MdTAC1a/b and plant branching types is also impressive. However, it seems not logical that the only differences in the MdTAC1a promoter sequence are associated with the branching types. The authors should show the data on the MdTAC1b promoter sequence in the F1 progeny.

Response 2: Thank for the suggestion. It is well known that plant branching angle is a quantitative trait, and controlled by QTLs. Four QTLs for weeping have been found in apple ( Dougherty et al., 2018), but not inclusive MdTAC1a/b, which indictes different gene regulates varied aspects of branching. Compared to MdTAC1a, variation in MdTAC1b promoters was complicated in four-type apple cultivars. A SNP ( at site of -158 bp) and a deletion of 1 bp ( at site of -47 bp) in its promoter were detected in colunar apples in comparition with spur, standard and weepping ciltivars, but other SNPs varied in four-type trees (Fig. 5 ). Thus, we start with MdTAC1a, and plan to develop SNP markers specific to MdTAC1b promoter in next work.

Point 3: In discussion section, the absence of the GAG motif is discussed with light response. However, any experimental evidence on relationship between the promoter sequences and light response is provided in the manuscript. The light condition indeed affect the MdTAC1a expression?

Response3: Thanks for the suggestion. The 6 bp (GAGAGA) in MdTAC1a promoter is predicted by PlantCare(http://bioinformatics.psb.ugent.be/webtools/plantcare/html/) as a light-responsive element GAG-motif. However, the relationship between the promoter sequences and light response need further experiments to explain.  One light-responsive element(GT1-motif) and three light-responsive element (G-box, GT1-motif and TCT-motif) were also found in the CDS of MdTAC1a/b. Overexpression of TAC1a/b displayed more sensitive to light response by bending stem with a larger curve in transformants compared dark condition (Fig.4). This phenomenon is reported in TAC1 transformation in Arabidopsis (Jessica et al., 2018).  Please see a detailed discussion in lines 432-438.

Point 4: Physical interaction between MdTAC1a and MdSRC2 seems informative. In the results of MdSRC2 experiments, it is necessary to provide more information for characterization of MdSRC2 protein since a number of C2 domain containing proteins exist in plants. “MdSRC2” came as bolt out of the blue.

Response4: Thanks for your proposal. Accoding to previous report, only one article report the function of SRC2 (Takahashi and Shimosaka,1997). Thus, we performed qRT-PCR to analyse expressions of MdSRC2 in the parental and F1 population, and found that it was highly expressed in apple trees with large branching angles. Please see detailed informantion in Result (lines348-351) and Dicsussion (lines 471-475)

Point 5: In page 9, the first paragraph, the last sentence, “these results indicate that the C-terminus of MdTAC1a binds to MdSRC2” is not explained by the provided data. What experiment indicates the importance of the C-terminus in terms of binding to MdSRC2?

Response5: Sorry for unclear discription.. In yeast two-hybrid system, the target gene is used as a bait protein, and a two-hybrid cross is performed in a yeast library to call out the protein that interacts with it. please see lines568-575

Point 6: Moreover, it is not tested in the manuscript at all that interaction between MdTAC1a and MdSRC2 is involved in regulation of branching angles. The last sentence in Abstract should be reconsidered.

Response 6: Thanks for the reminder. We revised the abstract in new version of this manuscript.

Point 7: Cellular localization of MdTAC1a and MdTAC1b is observed in tobacco. I think more careful interpretation to the observation is required. Especially GFP signals outside of the nucleus seems not only on the plasma membrane but in the cytosol. To exclude the possibility of cytosol localization of the GFP-fusion MdTAC1a/b, the authors may need to add some control experiments. In the provided data, “Control” seems to express free GFP. The free GFP can be accumulated in nucleus and cytosol.

Response 7: Sorry for the fuzzy photos. We replaced clear images in the revised manuscript.

Point 8: The legend in figure 1 indicates that “Control” is empty vector pCambia1300, but “Control” obviously showed GFP signal suggesting “Control” has 35S::GFP.

Response 8: Thank you for your suggestion. We corrected the mistake in the revised manuscript

Point 9: In BiFC experiment, YFP signals seems to be detected outside of the nucleus.

Response 9: Sorry for that. We replace clear images in the revised manuscript.

Point 10: Overexpression of MdTAC1a or MdTAC1b in tobacco provides nice results. Phenotypic differences between MdTAC1a OE plants and MdTAC1b OE plants are interesting, In the OE-plants, is the expression levels of MdTAC1a or MdTAC1b changed in light or dark condition? In other words, do the authors explain phenotypic differences between MdTAC1a OE plants and MdTAC1b OE plants as expression changes in light or dark condition even when the transgenes are under control of the 35S promoter?

Response 10: Thank you for suggestions. We added expression analysis of MdTAC1a/1b in MdTAC1-OE plants under light or dark conditions in the revised manuscript (Fig. 4F). The result showed that expressions of MdTAC1a/b increased significantly in MdTAC1a / MdTAC1b-OE plants under the control of 35S promoter in the light. And previous studies also demonstrated that light can affect the expression of TAC1.

Jessica, M.W.; Chris, D. TILLER ANGLE CONTROL 1 modulates plant architecture in response to photosynthetic signals. Journal of Experimental Botany 2018, 69, 4935–4944.

Point 11: In page3, comparison in expression levels between the transgenic plants and WT plants seems weird since WT plants do not contain the transgene (MdTAC1a or MdTAC1b).

Response 11: In the context of transformation, wild type plants were used as nagative control. Comparion in gene expression levels between the transgenic plants and WT plants is to confirm that target genes were transferred into plants.

In addition, all suggestions proposed by the reviewer have been revised in new version of manuscript.

Reviewer 3 Report

Manuscript ID: ijms-1568608

After careful review of the ms entitled “Characterization of MdTAC1a, related to branch angle in apple (Malus domestica Borkh.)” (Manuscript ID: ijms-1568608), this reviewer recommends it for publication after the suggested revisions.

This work investigated the TILLER ANGLE CONTROL1 (TAC1) genes, belonging to the IGT family, for the branch angle regulation of apple tree (Malus x domestica Borkh.). The design of the research was well structured. Initial analysis located MdTAC1a/b into cell compartments. Then, over-expression of MdTAC1a/b in tobacco enlarged leaf angles. Moreover, key genes for hormone synthesis, signalling, and photosynthesis were up-regulated in MdTAC1a/b transgenic plants. Gene expression analysis by qRT-PCR of MdTAC1a/b showed high expression in shoot tips and vegetative buds of weeping apple cvs, and weak expression in columnar cvs. InDel were observed in the MdTAC1a promoter (loss of 2 nt in spur and 6 nt in weeping cvs as compared with standard and columnar cvs), while several SNP were detected in the MdTAC1b promoter. A molecular marker specific for the InDel of MdTAC1a promoter was also developed to distinguish apple cvs and a F1 population. Y2H, BiFC, and Co-IP assays indicated that MdTAC1a interacts with MdSRC2, a protein related to intracellular calcium ion regulation by cold (from soy). In conclusion, differences in the MdTAC1a promoter are important for branch angle variation in apple and the interaction of MdTAC1a protein with MdSRC2 can regulate the expression of this character.

However, Authors have to exactly indicate the novelty of this study compared to other works on plant TAC1 (InDel and SNP of MdTAC1a/b promoters were already published for some apple cvs). At last, this reviewer suggests important revisions of the text, also by improving English language, and above all by removing inaccuracies.

Further suggestions are listed below, in the main text there were no line numbers:

TITLE

This reviewer suggests a new title: “Functional characterization of MdTAC1a gene related to branch angle in apple (Malus x domestica Borkh.)”

INTRO

Page 1

Line 1: insert “x” between “Malus” and “domestica

Line 3: line: remove “type” after “spur”

Line 11: line: explain acronym TAC1, see 1st line of abstract

Page 2

Line 23: move “however” to the beginning of this sentence

Line 27: check for tobacco plant: Nicotiana tabacum

Line 32: explain acronym SOYBEAN GENE REGULATED BY COLD 2 (SRC2) and insert a new reference to:

Takahashi, R.; Shimosaka, E. cDNA sequence analysis and expression of two cold-regulated genes in soybean, Plant Science 1997, 123(1–2): 93-104. https://doi.org/10.1016/S0168-9452(96)04568-2.

RESULTS

Par 2.1

Line 39: change “thus far” to “so far”

Page 3

Figure 1: bars of the measure units are not always clearly visible on the images

Page 4

In the text used for the description of Figure 3, in the first and second paragraph of page 4: there is not complete correspondence between times of Figure 3A (with 0, 1, 5, 12, 24 h) and 3C (with 0, 1, 3, 6, 8, 10, 12, 24 h); there is not complete correspondence between times of Figure 3B (with 0, 1, 8, 12, 24, 48 h) and 3D (with 0, 1, 3, 6, 8, 10, 12, 24, 48 h), please fix in the text of page 4 and in the Figure 3

In the first and second paragraph of page 4: there are many discordances, change Fig. 5B to Fig. 3B, please verify “In the OE plants, no bending was observed in the first 6 h.”, “WT and MdTAC1b-OE plants began to bend slightly upward after 3 h, whereas MdTAC1b-OE plants began bending slightly upward after 8 h (Fig. 3B).”

please verify “In addition, little difference was observed in the bending rate of MdTAC1a-OE and MdTAC1b-OE plants under light, whereas the bending rate of MdTAC1b-OE plants was significantly higher than that of MdTAC1a-OE in the dark.”, there are opposite trends under light and in the dark, differences are at the end (at 48 h) in the dark and no differences at 24 h under light, please rewrite

Page 5

In the caption of Figure 3: insert a description for letters C, D

Par. 2.3 and caption of Fig. 4: insert the time of sampling for gene expression analysis

Page 6

Par. 2.4

Line 1: please check the ref. Wang et al. (2018), it is [42], add ref. [20] and, consequently, change numbering of ref.

Page 7

In the caption of Fig. 5: change “differences” to “expression”, verify letters A and B, insert letters C and D.

In Figure 5C, verify numbering -275/-270, in the text it is -267/-273 (Line 4); however, to have six nt, it has to be from -267 to -272

Line 3: remove “Obvious”

Line 4: insert Figures 5C and 5D

Line 6: verify -267/-269, for two nt it has to be from -267 to -268

Line 7: insert Figure 5C for InDel and Figure 5D for SNP

Line 13: please verify and fix, change “Columnar- and spur-type” to “Standard-type”

Line 14-15: please verify and fix, change “Standard-type” to “Columnar-type”

Line 15: change Fig. 5B to Fig. 5D

Line 15-16: remove “by these examples” and “obviously”, insert “promoters” after MdTAC1a/b

Line 17: insert “primers” before “pair”, move “molecular-marker” and add “with” before 370-bp

Page 8

In Figure 6H, the numbering of gels is not legible

Line 1: use “primer pair”, not “pair of molecular markers”, or use “this molecular marker” without “pair”

Page 9

Par. 2.6: explain why Authors used MdTAC1a and excluded MdTAC1b

Line 1: change “cDNA” to “interaction”

Figure 7: describe in order Figure 7A (Y2H), 7B (Co-IP), and 7C (BiFC)

Explain acronyms the first time they are cited: BD, AD, BiFC, Co-IP etc.

Page 10

Caption of Figure 7: verify tobacco (N. tabacum) and fix; change SRC2 to SRC2

Par. 3.2, Line 12: remove “obvious”

Page 11

Par. 3.4, Lines 1-4, insert one or more reference(s)

L6: check for “amyloid deposition”, what do you mean by it ?

L8: check for “endothelial cells”, do you mean “endodermis cells” ?

L14: change GA to Ca

Page 11-12

Figure 8 it not clear, but the idea is good, please explain and describe better Figure 8, or remove it; what do you mean by ProMdTAC1, ProMdTAC1(delta) and “More MdTAC1a”? use a complete legend; Calcium is not correctly located, gravity has to be in both plants, what is the role of ARF2/3 PHOT1/2, etc. ?

Page 12

Par. 4.1, L1: insert the Latin name for apple (Malus x domestica Borkh.)

Par. 4.2, L1: what does genomic DNA refer to ? please, indicate the plant species

Par. 4.3, L1: what does cDNA refer to ? please, indicate the plant species

Par. 4.1, L7: insert the Latin name for tobacco (N. tabacum)

Page 14

  1. Conclusions

Line 3: change “an obvious” to “a”

Page 15

Change “Reference” to “References”

There are several typing errors in the References, and Authors lacking, please check and fix: for example, [1] Lespinasse, Y. followed by other Authors.

Supp. Figure S2

Change “element” to “elements”

Author Response

Response to Reviewer 3 Comments

Point 1: However, Authors have to exactly indicate the novelty of this study compared to other works on plant TAC1 (InDel and SNP of MdTAC1a/b promoters were already published for some apple cvs).

Response 1: Thanks for the suggestion. In previous work of our laboratory (Wang, 2018), four cultivars representing four ideotypes of apple trees with different branching angles are used to analyze difference in CDS of MdTAC1a/b and their promoters. Interestingly, variation in the promoter region rahter than CDS of MdTAC1 in apples probably influence tree form and branch angles in apple, absolutely different mechanism from other TAC genes in plants (Please see Discussion, lines448-454 ). In present study, 11 apple cultivars (Table 1) were used to clone the promoter region and CDS of MdTAC1a/b to prove the hypothesis. An IndDel marker specific to the pormoter of MdTAC1a corelated to variation of branch angles in 23 apple cultivars and 62 offsprings in a F1 population.

Point 2: At last, this reviewer suggests important revisions of the text, also by improving English language, and above all by removing inaccuracies.

Response 2: We apologize for this and revised the manuscript thoroughly.

Point 3: This reviewer suggests a new title: “Functional characterization of MdTAC1a gene related to branch angle in apple (Malus x domestica Borkh.)”

Response 3: Thanks for the seggestion. We have changed the title.

In addition, all suggestions proposed by the reviewer have been revised in the revised manuscript.

Round 2

Reviewer 1 Report

Abstract

L-18-20 Sentence is incomplete, please revise it.

Abbreviations used in the abstract need to define at first instance for (IGT, GA2ox, Y2H, BiFC Co-IP).

Introduction:

L69-70 poplar “(Populus × zhaiguanheibaiyang)” this cannot be the scientific name of poplar. Mentioned name is for a hybrid. Please write the correct scientific name for poplar.

 L72-74 “No difference in cDNA differences but variation in promoters of two genes were detected among analyzed ideotypes, but variation was observed in the promoters [20]” This sentence is wrong please revise as below.

“There was no difference in cDNA however variation in the promoters of two genes among the examined ideotypes was identified [20]”

L-94-95 “The MdTAC1a protein was detected in the cell nucleus and cell membrane, whereas the MdTAC1b protein was localized to the cell membrane (Fig. 1)”.

Previously I said when you are writing the protein name of “MdTAC1a and b” it should not be italic so write it as “MdTAC1a” and “MdTAC1b” and do not remove protein word keep as it is. Similarly, in Figure 1. legends do not italic the “MdTAC1a” and “MdTAC1b”

Revise subheading: as “2.2. Phenotypic and key genes analysis of transgenic MdTAC1a/b in tobacco”

Fig. 3. There is still a lack of statistical significance on the bar graphs please revise it replace the figure with statically significance deference either with an asterisk or different letters. Follow the same comment for Figure 4. C-F. and remaining all figures should show statically significance indication on respective bars where significant difference is observed.

3.4. The effect of MdSRC2–MdTAC1a interaction on plant Ca2+concentration

Cannot start sentence with SRC2…….. please revise it write it appropriately.  

Conclusion:

L604 “hybrid F1 progeny” please write as “F1 Hybrid”

Please correct the spelling mistake of “development”

“F1 offspring” just use offspring “F1 offsprings” F1 is a hybrid

Author Response

Response to Reviewer 1 Comments

Point1: Moderate English changes required

Answer 1: We apologize unreservedly for many grammatically incorrect expressions throughout the entire manuscript. Those incorrect writing has been amended in new version. In fact, our manuscript was technically edited by the Edanz's editing services, and it was Chinese New Year, the relevant certification could not be issued. If necessary, we will attach the certificate when we get it.

Point 2:L69-70 poplar “(Populus × zhaiguanheibaiyang)” this cannot be the scientific name of poplar. Mentioned name is for a hybrid. Please write the correct scientific name for poplar.

Answer 2: We checked the cited article, in which it is stated as follows, so we cited the Latin names according to the article.

‘Here, we cloned two genes from Populus × zhaiguanheibaiyang (a narrow-crown poplar), designated PzTAC and PzLAZY, which were predicted to be members of the ITG gene family through sequence homology.’

Point3: Fig. 3. There is still a lack of statistical significance on the bar graphs please revise it replace the figure with statically significance deference either with an asterisk or different letters. Follow the same comment for Figure 4. C-F. and remaining all figures should show statically significance indication on respective bars where significant difference is observed.

Answer 3: Thanks to the reviewer's suggestion, we have added ANOVA to the figures in the revised manuscript.

In addition, all suggestions for writing proposed by the reviewer have been revised in new vesion of the manuscript.

Reviewer 2 Report

Dear the editor,

The revised manuscript has been rewritten according to the comment.

I think I have no additional comments on the manuscript.

Best

Author Response

Response to Reviewer 2 Comments

Dear Reviewers, Thank you for some constructive suggestions you have provided in our manuscript.

Response to Reviewer 1 Comments

Point1: Moderate English changes required

Answer 1: We apologize unreservedly for many grammatically incorrect expressions throughout the entire manuscript. Those incorrect writing has been amended in new version. In fact, our manuscript was technically edited by the Edanz's editing services, and it was Chinese New Year, the relevant certification could not be issued. If necessary, we will attach the certificate when we get it.

Point 2:L69-70 poplar “(Populus × zhaiguanheibaiyang)” this cannot be the scientific name of poplar. Mentioned name is for a hybrid. Please write the correct scientific name for poplar.

Answer 2: We checked the cited article, in which it is stated as follows, so we cited the Latin names according to the article.

‘Here, we cloned two genes from Populus × zhaiguanheibaiyang (a narrow-crown poplar), designated PzTAC and PzLAZY, which were predicted to be members of the ITG gene family through sequence homology.’

Point3: Fig. 3. There is still a lack of statistical significance on the bar graphs please revise it replace the figure with statically significance deference either with an asterisk or different letters. Follow the same comment for Figure 4. C-F. and remaining all figures should show statically significance indication on respective bars where significant difference is observed.

Answer 3: Thanks to the reviewer's suggestion, we have added ANOVA to the figures in the revised manuscript.

In addition, all suggestions for writing proposed by the reviewer have been revised in new vesion of the manuscript.